

# Characteristics of the soil microbial community in the forestland of *Camellia oleifera*

Peng Zhang[1,2,*], Zhiyi Cui[2,*], Mengqing Guo[1] and Ruchun Xi[1,3]

[1] College of Forestry and Landscape Architecture, South China Agricultural University, Guangzhou, China
[2] Research Institute of Tropical Forestry, Chinese Academy of Forestry, Guangzhou, Guangdong, China
[3] The State Key Laboratory for Conservation and Utilization of Subtropical Agro–Bioresources, Guangzhou, China
[*] These authors contributed equally to this work.

## ABSTRACT

Characterizing soil microbial community is important for forest ecosystem management and microbial utilization. The microbial community in the soil beneath *Camellia oleifera*, an important woody edible oil tree in China, has not been reported before. Here, we used Illumina sequencing of 16S and ITS rRNA genes to study the species diversity of microorganisms in *C. oleifera* forest land in South China. The results showed that the rhizosphere soil had higher physicochemical properties, enzyme activities and microbial biomass than did the non-rhizosphere soil. The rhizosphere soil microorganisms had a higher carbon source utilization capacity than the non-rhizosphere soil microorganisms, and attained the highest utilization capacity in summer. The soil microbial community of *C. oleifera* was characterized by rich ester and amino acid carbon sources that played major roles in the principal functional components of the community. In summer, soil microbes were abundant in species richness and very active in community function. Rhizosphere microorganisms were more diverse than non-root systems in species diversity, which was associated with soil pH, Available phosphorous (AP) and Urease (URE). These results indicated that microbial resources were rich in rhizosphere soil. A priority should be given to the rhizosphere microorganisms in the growing season in developing and utilizing soil microorganisms in *C. oleifera* plantation. It is possible to promote the growth of *C. oleifera* by changing soil microbial community, including carbon source species, pH, AP, and URE. Our findings provide valuable information to guide microbial isolation and culturing to manage *C. oleifera* land.

# INTRODUCTION

Photosynthetically fixed compounds are released into the soil, especially rhizosphere soil, through root exudates and plant residues by green plants (*Hütsch, Augustin & Merbach, 2002*; *Shah et al., 2016*). These compounds are the energy and carbon sources of soil microorganisms. Soil microorganisms, in turn, exploit inorganic, organic and atmospheric

Corresponding author
Ruchun Xi, xirc2006@scau.edu.cn

forms of nutrients for plant growth (*Shah et al., 2016*). Microorganisms participate in biochemical processes such as soil ammoniation and nitrification, by promoting soil organic matter decomposition and nutrient conversion, and they are the primary drivers of biochemical cycles (*Moreau et al., 2019*). They play extremely important roles in the formation of soil fertility and the transformation of plant nutrients.

Microbial community structure characteristics refer to the species composition and abundance of microbial communities (*Ochoa-Hueso et al., 2018*). An increase in the soil microbial community structure and composition diversity helps to maintain the stability of the soil ecosystem and enhances the resistant capacity of soil against environmental degradation (*Chen et al., 2017*). Studies has been made toward the characterization of soil microbiomes in some plant species, including *Arabidopsis* (*Bulgarelli et al., 2012*), rice (*Edwards et al., 2015*), *Eucalyptus saligna* (*Curlevski et al., 2014*), and *Populus* (*Gottel et al., 2011*).

*Camellia oleifera* (*C. oleifera*), a unique edible oil tree species in China, is one of the world's four famous woody oil plants (*Jia et al., 2015*; *Zhou et al., 2019*). Its unsaturated fatty acid content is higher than 90% (*Wu et al., 2019*); this species have strong anti-oxidation and free radical scavenging ability and the effect of preventing cardiovascular diseases. The history of *C. oleifera* cultivation in China is over 2300 years (*He et al., 2011*), and the current areas of cultivation in China are predicted to grow to 4.67 million ha by 2020 (*Liu et al., 2017*). *C. oleifera* is not only an excellent economic tree but also an excellent ecological tree (*Wang, Kent & Fang, 2007*). *C. oleifera* generally grows in the mountains and hills of subtropical regions in southern China and is also of great value in soil and water conservation and maintaining ecological balance.

The method of laboratory culture has been used to study soil microorganisms of *C. oleifera* (*Liu et al., 2017*; *Liu et al., 2018*; *Lu, Shen & Chen, 2019*). These studies revealed the number and types of soil microbial communities in *C. oleifera* in different seasons. However, this approach lacks precision, and many community characteristics had not been explored. Instead, high throughput sequencing can be more powerful to investigate soil microbial community structure. To our knowledge, high-throughput sequencing methods have not been used in the study on the characteristics of the soil microbial community in *C. oleifera* forest land. Obtaining such information in the production and management of the *C. oleifera* forest ecosystem will be valuable in soil management and microbial utilization in the future. Due to rhizosphere effect, there are more microorganisms in rhizosphere soil than in non-rhizosphere soil. Rhizosphere microbes, which are considered to be part of the second genome of plants, play a fundamental role in plant growth and health. We hypothesized that *C. oleifera* have a "rhizosphere effect", and the nutrient characteristics of rhizosphere soil is better than non-rhizosphere soil.

## MATERIALS & METHODS

### Site description

The research area is located in Fengren, Longchuan, Heyuan, Gangdong, China (24°13′38″N, 115°20′18″E). The average altitude is 154 m. The average annual temperature

is 21.8 °C, and annual rainfall is 1,501.8 mm (http://www.tianqi.com/qiwen/city_longchuan/, in Chinese). The soil is categorized as haplic acrisols according to the FAO soil classification. The *C. oleifera* forest was cultivated in the early 1960s, and there is no artificial disturbance except the annual harvest. The height of forest is $(3.71 \pm 1.08)$ m, the crown diameter is $(4.47 \pm 1.15)$ m, and the ground diameter is $(8.23 \pm 3.49)$ cm.

Sample collection

Six 20 m × 20 m sample plots were randomly set in the experimental area. In April 2018 (spring), July 2018 (summer), October (autumn) and January 2019 (winter), soil samples were collected at each sample site under sunny conditions.

Fifteen healthy *C. oleifera* trees were selected from each plot. The rhizosphere and non-rhizosphere soil samples were collected as described by *Xu et al. (2018)*. The rhizosphere and non-rhizosphere soil samples from the same grove were mixed together as one sample for each plot. The top 5-cm of soil was removed, and fine roots (approximately 1 mm diameter) from a depth of 5–15 cm were collected. The roots were removed from the soil with a shovel and then gently shaken to remove the soil that was not tightly attached to the roots. The rhizosphere soil was obtained by gently brushing the soil on the root surface with a sterile brush. Soil collected from the same 5–15 cm depth in locations between trees and without roots was termed non-rhizosphere soil. The samples were stored in liquid nitrogen and transported to the laboratory for refrigeration.

## Measurement methods
### Soil property analysis

Soil texture was measured as described by *Will et al. (2010)*. The pH was measured in soil of water suspension with a ratio of 1:2.5 using glass electrode pH meter; the content of organic matter (OM) was determined by the high-temperature external heat dichromate oxidation-capacity method (*Schinner et al., 1995*); alkali hydrolysis (AN) was used in the determination of alkali hydrolysis nitrogen by the alkali-hydrolysis diffusion method (*Ming, Klemens & Zhang, 2011*); and the available phosphorous (AP) was extracted by ammonium hydrochloride and determined by the molybdenum antimony colorimetric method (*Olsen & Sommers, 1982*).

Cellulase (beta-glucosamine) (CEL) activity was determined using the salicylic glycosides colorimetric method (*Miller, 1959*); urea enzyme (URE) was determined using the sodium phenol colorimetric method (*Bell et al., 2013*); and acid phosphatase (ACP) was determined using the benzene diphosphate colorimetric method (*Lu, 2000*).

Soil microbial biomass carbon (MBC), nitrogen (MBN) and phosphorus (MBP) were extracted by chloroform fumigation (*Vance, Brookes & Jenkinson, 1987*; *Petersen, Petersen & Rubæk, 2003*).

### Soil microbial functional diversity analysis

Soil microbial metabolic activity was measured using Biolog Eco-microplates (*Garland, 1997*). Carbon utilization by microorganisms caused colour development in the wells. The colour development in each well was recorded at an optical d at 24 h intervals for 168 h. The different carbon sources of soil microorganism utilization were analysed in 120 h.

### Soil microbial community characteristics analysis

The 16 s rRNA V3-V4 region of bacteria and the internal transcribed spacer (ITS) region of fungi were used as the target sequences of DNA. The PCR amplification was carried out with the universal primer pairs of 338F (5′- ACTCCTACGGGAGCAGCAG -3′) and 806R (5′-GGACTACHGGG TWTCTAAT-3′), and ITS1F (5 ′- CTTGGTCATTTAGGAAGTAA-3′) and ITS2R (5′- GCTGCGTTC TTCATCATGATGC -3′), respectively (*Biddle et al., 2008*; *Mukherjee et al., 2014*). The 20 µL PCR amplification reaction material comprised of 4 µL of 5× FastPfu Buffer (16 s v3-v4)/2 µL of 10× Buffer (ITS), 2 µL dNTPs (2.5 mM), 0.8 µL each of forward and reverse primers, 0.4 µL of FastPfu Polymerase (16S v3-v4)/0.2 µL of rTaq Polymerase (ITS), 0.2 µL of BSA, and 10 ng template DNA. Each of these was supplemented with ddH$_2$O up to 20 µL

Polymerase chain reaction (PCR) amplification followed a specific thermal program: an initial denaturation at 95 °C for 3 min, followed by 28 cycles of denaturation at 95 °C for 30 s, annealing at 55 °C for 30 s, and elongation at 72 °C for 45 s, and a final extension at 72 °C for 10 min. Three repeated PCR products from the same sample were mixed and detected by 2% agarose gel electrophoresis. The samples with bright main bands between 400 bp and 450 bp were selected for further analysis. The PCR products were recovered and purified by the AxyPrep$^{TM}$ DNA Gel Extraction Kit (Axygen Biosciences, Union City, CA, USA) and quantified using a QuantiFluor®-ST Fluorometer (Promega, USA). Finally, the Illumina pair-end library preparation, cluster generation and 250 bp pair-end sequencing were determined by Majorbio Bio-Pharm Technology Co. Ltd. (Shanghai, China).

## Data analysis

QIIME v1.7.0 was used to depolymerize, quality filter and analyse the raw Illumina sequences and to obtain high quality tags data. After comparison with GOLD (Genomes Online Database), the detected chimera sequence was removed to obtain valid tags data. The sequence obtained was clustered into operational taxonomic units (OTUs) according to the similarity level using UPARSE Pipeline v 7.0 at a threshold of 97% classification. The representative sequences in OTUs were then selected and compared with the known sequence in the database to obtain the species annotation information (*Zhang et al., 2019*).

SPSS Statistics 22.0 was used for one-way ANOVA, multiple comparisons, and principal component analysis. Hierarchical clustering was used to assess similarity characteristics soil based on the unweighted pair group arithmetic average (UPGMA). Unweighted Principal Coordinates Analysis (PCoA) of microbial community data was performed using the vegan package of R.

## RESULTS

### Soil physicochemical properties, enzyme activities and microbial biomass

In the four seasons, the rhizosphere soil pH was lower than that of the non-rhizosphere soil, and the OM and AN and AP contents of the rhizosphere soil were significantly higher than these of non-rhizosphere soil (Table 1). The pH of rhizosphere and non-rhizosphere

**Table 1 Physicochemical, enzyme activities and microbial biomass characteristics of the soil samples.**

| Item | Soil | Spring | Summer | Autumn | Winter |
|------|------|--------|--------|--------|--------|
| pH | R | 4.97 ± 0.19b | 4.94 ± 0.27b | 4.95 ± 0.87b | 5.16 ± 0.04a |
| | N | 5.56 ± 0.35a[*] | 5.20 ± 0.29b[*] | 5.33 ± 1.14b[*] | 5.35 ± 0.06b[*] |
| OM (g kg$^{-1}$) | R | 26.56 ± 6.08c[*] | 34.24 ± 6.11a[*] | 31.22 ± 5.87b[*] | 23.36 ± 1.48d[*] |
| | N | 7.61 ± 4.05c | 15.20 ± 5.06a | 8.23 ± 0.86b | 6.67 ± 2.94d |
| AN (g kg$^{-1}$) | R | 88.71 ± 22.04d[*] | 116.51 ± 34.67b[*] | 129.91 ± 29.47a[*] | 102.25 ± 28.54c[*] |
| | N | 32.16 ± 22.94d | 73.08 ± 42.05b | 80.39 ± 26.21a | 53.81 ± 14.87c |
| AP (g kg$^{-1}$) | R | 2.33 ± 3.75c[*] | 3.21 ± 0.91b[*] | 3.96 ± 3.24a[*] | 2.23 ± 0.02c[*] |
| | N | 1.08 ± 0.05c | 1.70 ± 0.08b | 2.06 ± 0.06a | 0.54 ± 0.01d |
| Cellulase (μmol g$^{-1}$ h$^{-1}$) | R | 0.28 ± 0.03c[*] | 0.47 ± 0.04a[*] | 0.39 ± 0.05b[*] | 0.25 ± 0.04c[*] |
| | N | 0.08 ± 0.07b | 0.28 ± 0.04a | 0.26 ± 0.09a | 0.07 ± 0.01b |
| Urease (μmol g$^{-1}$ h$^{-1}$) | R | 10.50 ± 4.04b[*] | 15.11 ± 2.41a[*] | 16.52 ± 2.64a[*] | 7.40 ± 0.26c[*] |
| | N | 7.73 ± 4.58b | 8.22 ± 2.75a | 9.14 ± 3.58a | 7.00 ± 0.02b |
| Acid phosphatase (μmol g$^{-1}$ h$^{-1}$) | R | 6.10 ± 2.18c[*] | 9.84 ± 0.86a[*] | 7.76 ± 1.24b[*] | 5.79 ± 1.40d[*] |
| | N | 3.56 ± 0.41b | 5.03 ± 0.55a | 4.37 ± 0.57a | 3.06 ± 0.26b |
| MBC (mg kg$^{-1}$) | R | 127.23 ± 31.56d[*] | 268.54 ± 41.35a[*] | 232.23 ± 38.76b[*] | 159.23 ± 26.54c[*] |
| | N | 68.17 ± 12.47d | 115 ± 24.86a | 101 ± 20.63b | 81.3 ± 15.85c |
| MBN (mg kg$^{-1}$) | R | 31.24 ± 8.56c[*] | 52.24 ± 12.27a[*] | 44.23 ± 10.57b[*] | 18.53 ± 6.52d[*] |
| | N | 15.31 ± 2.54c | 28.25 ± 6.21a | 21.24 ± 8.55b | 10.37 ± 2.33d |
| MBP (mg kg$^{-1}$) | R | 1.57 ± 0.07b[*] | 3.16 ± 0.08a[*] | 2.92 ± 0.04a[*] | 1.12 ± 0.32c[*] |
| | N | 0.94 ± 0.05b | 1.63 ± 0.15a | 1.15 ± 0.93b | 0.62 ± 0.02c |

**Notes.**

[*]indicates that the difference between the different treatments in the same season is significant at the 0.05 level. Different letters indicate that the same treatment differs significantly at the 0.05 level in different seasons.

R, rhizosphere soil; N, non-rhizosphere soil; OM, organic matter; AN, alkali nitrogen; AP, available phosphorus.

soil of *C. oleifera* were the lowest in summer. The content of OM, AN and AP were the highest in summer or autumn.

Cellulase and acid phosphatase in rhizosphere and non-rhizosphere soils were most active in summer, whereas the highest activity of urease was observed in autumn. There were significant differences in soil enzyme activity between rhizosphere and non-rhizosphere soils in each season.

The highest contents of microbial biomass carbon, nitrogen and phosphorus in the rhizosphere and non-rhizosphere soils were found in summer. There were significant differences in the microbial biomass content between rhizosphere and non-rhizosphere soils in each season.

## Soil microbial functional diversity
### Kinetic characteristics of soil microbial utilization of all carbon sources
By the time before 24 h, average well colour development (AWCD) values were very low, less than 0.051 (Fig. 1), which means that the microbial activity was weak, and the carbon source was basically not consumed. Between 24 h and 144 h, AWCD values increased rapidly, indicating that the carbon source was heavily utilized after 24 h. In the same season, the AWCD values of rhizosphere soil were higher than those of non-rhizosphere soil in each culture period, indicating higher carbon source utilization and stronger metabolic activity

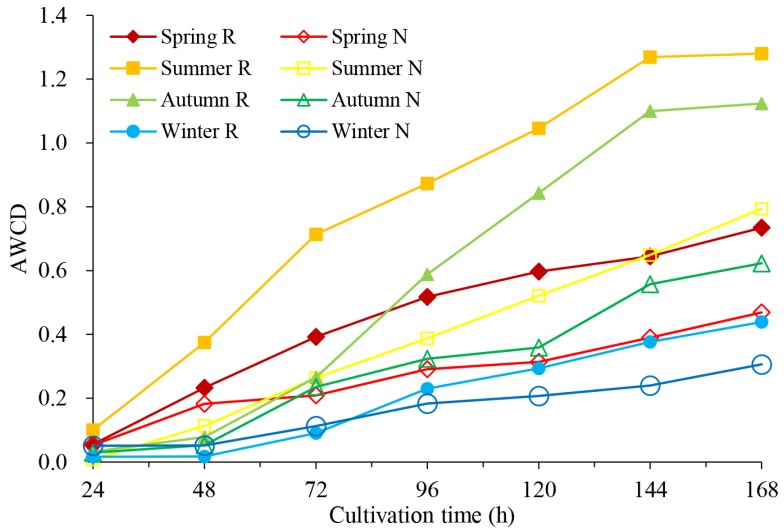

**Figure 1** Average well color development (AWCD) of soil microbial community in *C. oleifera* soil changes with culture time.

of the rhizosphere microbial community. Within one year, the carbon source capacity for the rhizosphere soil and the non-rhizosphere soil was consistent, from summer to autumn, spring and winter.

### Types of carbon sources used by soil microbial communities

In annual, six major types of carbon sources were utilized by rhizosphere soil microbial communities (Fig. 2). However, the average utilization of various carbon sources was different. In summer, the average utilization of various carbon sources in the rhizosphere soil microbial community was from esters, alcohols, acids, amines, amino acids and sugars. The average utilization degree of non-rhizosphere soil microbial communities with respect to various carbon sources was esters, acids, alcohols, amino acids, sugars and amines.

## Soil microbial community characteristics
### Sequencing data statistics

The number of effective sequences of bacteria and fungi were 29,570–63,373 and 31,824–65,373, respectively (Fig. 3). The curve gradually flattened, and the sequencing quantity tended to saturate. This indicates that the construction of the library was reasonable, and the sampling depth of bacteria and fungi was sufficient for the species diversity.

In total, 550 species of bacteria were detected in rhizosphere and non-rhizosphere soils, belonging to 26 families, 60 classes, 119 orders, 206 families, 297 genera. A total of 6 families, 23 classes, 71 orders, 139 families, 215 genera and 292 species of fungi were obtained.

## Sample comparative analysis

According to the hierarchical clustering tree and principal component analysis (Figs. 4 and 5), we knew that among rhizosphere soil and non-rhizosphere soil, the bacterial

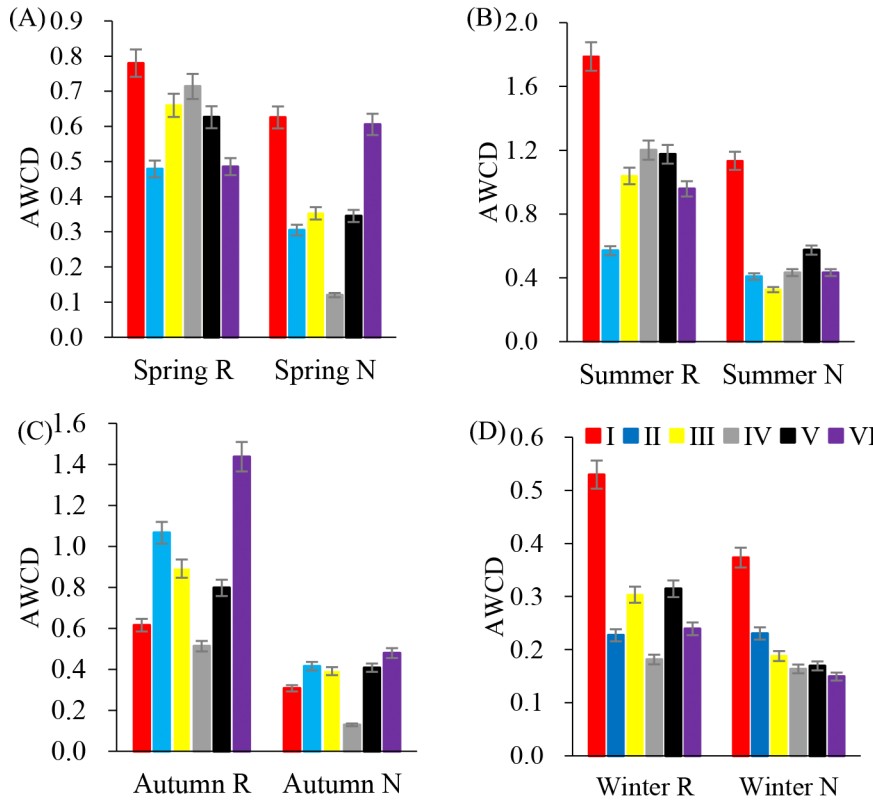

**Figure 2** **Substrates utilization pattern in soil microorganisms.** I (Esters); II (Saccharides); III (Amines); IV (Alcohols); V (acids); VI (Amino acids). (A) Spring; (B) Summer; (C) Autumn; (D) Winter.

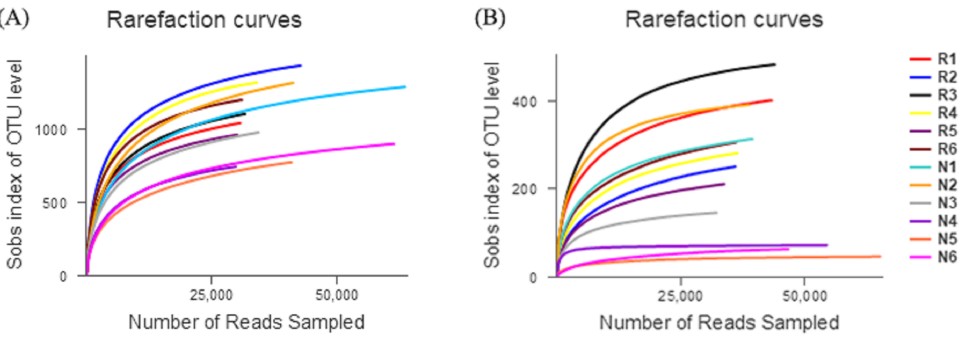

**Figure 3** **Rarefaction curves of different soil microbial communities.** (A) Bacterial community; (B) fungal community.

composition and structure of soil were very similar (Figs. 4A and 5A), and there were some differences in fungal composition between rhizosphere soil and non-rhizosphere soil (Figs. 4B and 5B). Especially, two outlier fungal communities (N5 and N6) were the outliers in the PCAs.

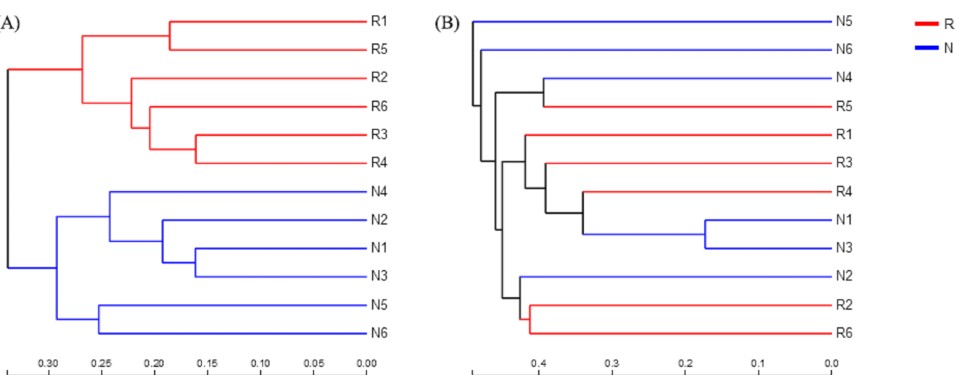

**Figure 4** **Hierarchical clustering tree of different soil microorganisms at OTU level.** (A) Bacterial community; (B) fungal community.

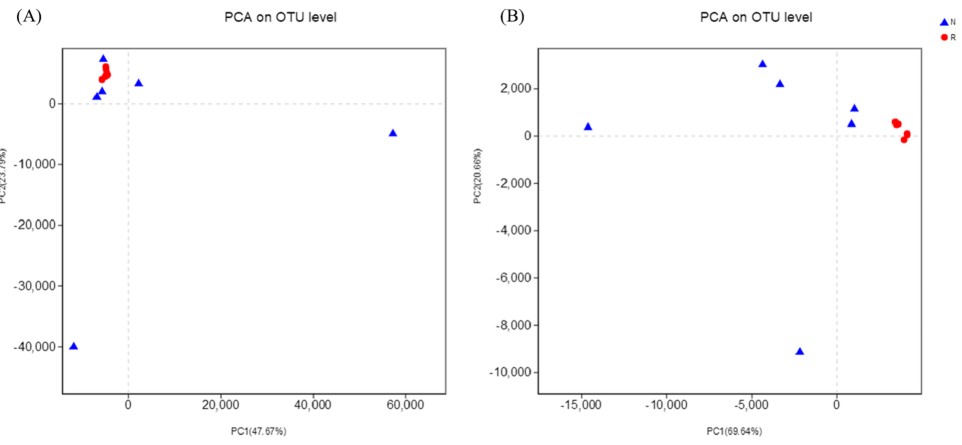

**Figure 5** **Principal component analysis of different soil microorganisms.** (A) Bacterial community; (B) fungal community.

## Species composition analysis
### Analysis of species Venn diagram

A total of 1936 OTUs were classified into bacterial communities (Fig. 6). Among them, the number of OTUs in rhizosphere and non-rhizosphere soils was 308 and 137, and the total number of OTUs in rhizosphere and non-rhizosphere soils was 1491 (Fig. 6A). Fungi were classified into 1,068 OTUs. Among them, the number of OTUs in rhizosphere and non-rhizosphere soil was 517 and 159, and the total number of OTUs was 392 (Fig. 6B).

## Community composition analysis
### Community structure at the phylum classification level

According to the Bar map of the community, we knew that at the phylogenetic level, soil bacteria in each sample belonged to 26 phyla, of which 12 phyla were found in rhizosphere and non-rhizosphere soils with a relative abundance higher than 1% (Fig. 7A). Soil fungi

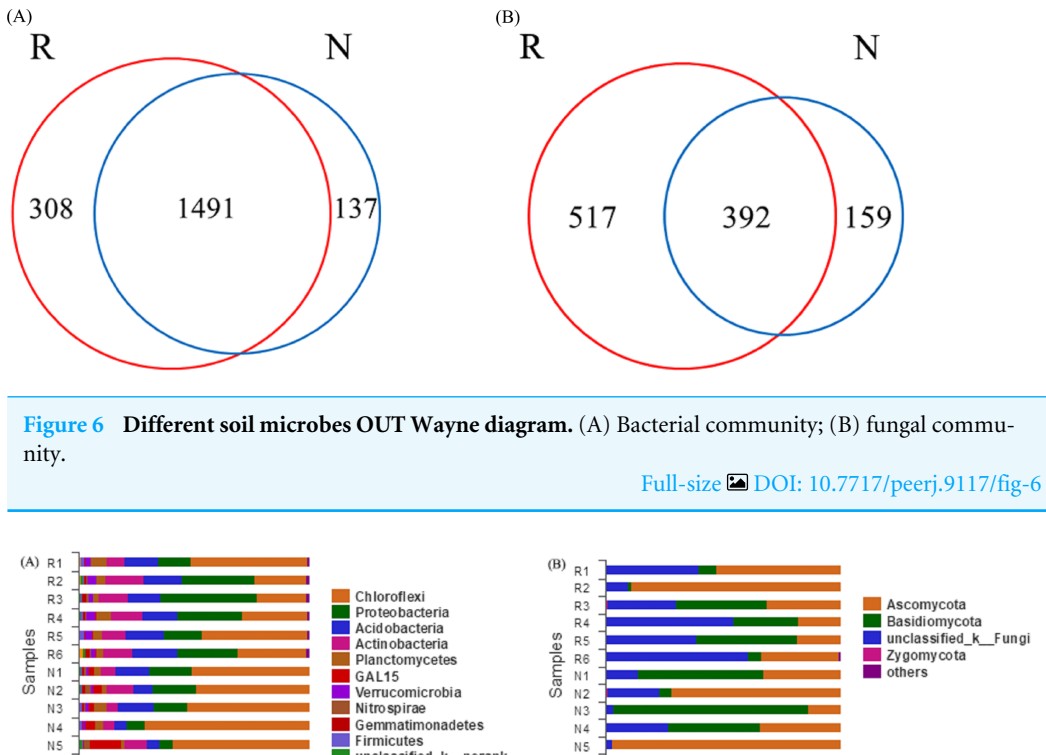

**Figure 6** **Different soil microbes OUT Wayne diagram.** (A) Bacterial community; (B) fungal community.

**Figure 7** **Abundance distribution of different soil microorganisms at the phylums level.** (A) Bacterial community; (B) fungal community.

in each sample belonged to 12 phyla, of which 4 phyla were found in rhizosphere and non-rhizosphere soils with a relative abundance higher than 1% (Fig. 7B).

### Community structure at the genus classification level

In total, 297 genera of bacteria were obtained from soil samples, and 50 genera were found in rhizosphere and non-rhizosphere soils with abundances higher than 1% (Fig. 8A). The 14 dominant species in the rhizosphere soil accounted for 53.62% of the total abundance. The 14 dominant species of non-rhizosphere soil samples accounted for 76.14% of the total abundances.

In total, 215 genera of fungi were obtained from soil samples, and 26 genera were found in rhizosphere and non-rhizosphere soils with abundances higher than 1% (Fig. 8B). The 8 dominant species in the rhizosphere soil accounted for 89.25% of the total abundance. The 5 dominant species of non-rhizosphere soil samples accounted for 89.64% of the total abundance.

### Correlation analysis of community structure and environmental factors

The environmental factors pH and AP were associated with the soil bacterial community (Fig. 9A). The higher the pH value is, the lower the bacterial community abundance would
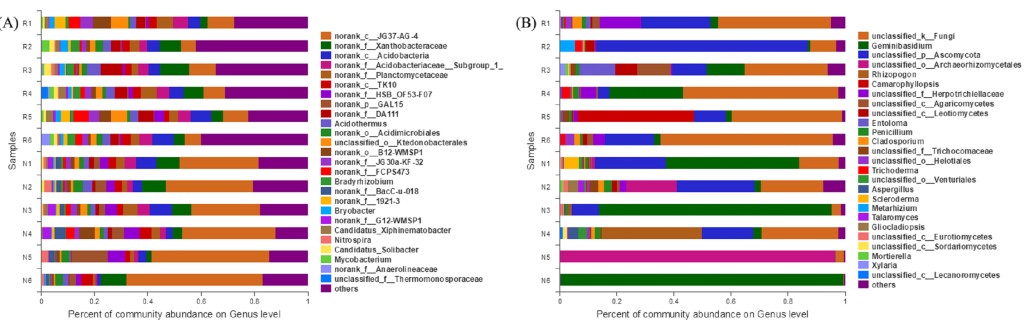

**Figure 8 Abundance distribution of different soil microorganisms at the genus level.** (A) Bacterial community; (B) fungal community.

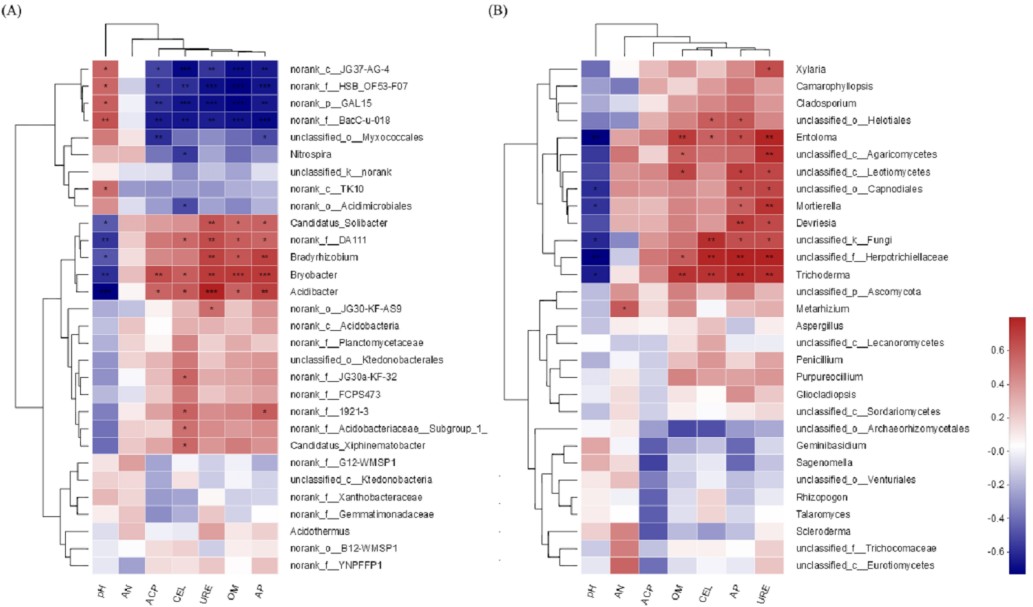

**Figure 9 Spearman correlation heatmap of soil microorganisms.** (A) Bacterial community; (B) fungal community. The corresponding value of heat map is Spearman corralation coefficient where "r" value is between −1 and 1, $r < 0$ is negative correlation, $r > 0$ is positive correlation. * indicates significance test $0.01 < P \leq 0.05$, ** indicates significance test $0.001 < P \leq 0.01$, *** indicates significance test $P \leq 0.001$.

be, and when the content of AP was the highest, the abundance of bacteria community was the highest. For the soil fungal community, the environmental factors pH and URE were associated with the soil fungal community (Fig. 9B). The higher the pH value is, the lower the fungal abundance would be, and the greater the URE activity is, the higher the fungal community abundance would be.

## DISCUSSION

*C. oleifera* is often planted in mountainous areas with poor site conditions, and fertilization and other management measures lag behind. Therefore, the study of soil nutrient

characteristics and soil microbial community characteristics of *C. oleifera* is of great helpful to the soil management of this species. Soil nutrient is significantly affected by soil microbes through mineralization and nutrient retention (*Zhang et al., 2018a*; *Zhang et al., 2018b*; *Shao et al., 2019*). In this study, the pH value in the rhizosphere soil of *C. oleifera* was lower than that in the non-rhizosphere soil, and the content of OM, AN and AP in the rhizosphere soil was significantly higher than that in the non-rhizosphere soil. This reflects the nutrient enrichment effect of *C. oleifera* rhizosphere, and *C. oleifera*, *Malus pumila*, *Pseudotsuga menziesii*, *Fagus sylvatica*, *Helianthus annuus* and other plants have a strong "rhizosphere effect" (*Dong & Shu, 2001*; *Fu & Cheng, 2004*; *Augustine et al., 2011*; *Calvaruso, N'Dira & Turpault, 2011*; *Dotaniya & Meena, 2015*; *De Feudis et al., 2016*). It is well known that soil biological (microbial and enzymatic) activities are highly correlated with soil physicochemical properties and also participate in the conversion of soil nutrients (*Gispert et al., 2013*). The activity of soil enzymes in the *C. oleifera* system was generally higher in summer and autumn and lower in winter during the dormant period. All biochemical transformations in soil are facilitated by enzymes; they can promote the transformation of soil nutrients and accelerate the soil nutrient cycle (*Hoang et al., 2016*). However, the growing season is the period in which plants have the greatest demand for nutrients. Additionally, MBC, MBN and MBP are usually regarded as indicators of microbial nutrient requirements and nutrient availability (*Dijkstra et al., 2012*; *Mooshammer et al., 2017*). C, N and P contents in microbial biomass were the highest in summer, and these results are in line with several previous findings (*Alongi, 1988*; *Bardgett et al., 1999*; *Chen et al., 2015*; *Zifčáková et al., 2016*; *Spohn et al., 2018*). Due to the presence of soil rhizosphere microorganisms, the soil quality in the rhizosphere of *C. oleifera* is superior to non-rhizosphere soil, which is more conducive to the growth of *C. oleifera*. In other words, it is of great significance for the development and utilization of the rhizosphere microorganisms of *C. oleifera*.

The pattern of Biolog metabolic diversity is related to microbial community composition and is sensitive to the change of the functional microbial community (*Rogers & Tate III, 2001*). The AWCD reflects the integrated capability of soil microbes to use a carbon source and the microbial activity (*Wang et al., 2016*). This study showed that the metabolic activity of the rhizosphere microbial community was stronger than that of non-rhizosphere soil microorganisms, and the use of carbon sources in summer was higher than that in the other seasons. Based on the utilization of six kinds of carbon sources, rhizosphere soil and non-rhizosphere soil microorganisms made greater use of ester and amino carbon sources. In production practice, understanding the functional of microbial community has positive significance for the soil management of *C. oleifera* during the four seasons.

Many microbial species are same in rhizosphere and non-rhizosphere soil, but some are different. Rhizosphere soil microbial community diversity was higher than that in non-rhizosphere soil. This also means that the ability of rhizosphere soil to resist the change of the external environment is stronger, and its soil ecological environment is more stable (*Guo et al., 2018*; *Xia et al., 2018*). Microbial community diversity of non-rhizosphere soil was lower than rhizosphere soil. Especially, there were very few fungal species in some samples. The probable reason may be non-rhizosphere soil lack of diverse substrates. And

this result was possible caused by experimental problems in sequencing. The reason of the formation in the forestland of *C. oleifera* deserve further studied.

*Chloroflexi, Proteobacteria, Acidobacteria, Actinobacteria* and *Planctomycetes* were the dominant groups of the bacterial community of *C. oleifera. Ascomycota, Basidiomycota*, etc., were the dominant groups of the fungal community of *C. oleifera*. The dominant species of *C. oleifera* soil have similarities with the soil of many woodlands (*Youssef & Elshahed, 2009; Frac et al., 2018; Prieto-Barajas, Valencia-Cantero & Santoyo, 2018; Egidi et al., 2019). Chloroflexi, Proteobacteria, Actinobacteria* and *Planctomycetes* bacteria can catalyze nitrification that promotes of the biogeochemical nitrogen cycle process (*Kong, Nielsen & Nielsen, 2005; Hayatsu, Tago & Saito, 2008; Sorokin, Lücker & Vejmelkova, 2012; Shridhar, 2012). Acidobacteria* bacteria is beneficial to plant growth in acid soil (*Yang, Liu & Li, 2012). Ascomycota*, and *Basidiomycota* fungal can form mycorrhiza in symbiosis with plants, and benefit to plant nutrition absorption and growth (*Toju, Yamamoto & Sato, 2013*). These dominant microbial groups play important roles in the function of *C. oleifera* soil.

The environmental factors pH and AP were associated with the soil bacterial community. For the soil fungal community, the associated environmental factors were pH and URE. During the last years, several studies targeting soil microbial communities came to the same conclusion that soil pH is the major driver of microbial communities (*Lauber et al., 2013; Kaiser et al., 2016*). A range of root exudates may alter the diversity and abundance of carbohydrate inputs into soil, as well as soil pH, AP and URE, which is an important driver of soil microbial community diversity and structure. It means that management of soil physical and chemical properties can affect microbial community structure. Also, this provides a direction for enhancing plant growth and productivity.

In summer, the growth season that is the most abundant in terms of microbial species, soil microorganisms have the greatest impact on plant nutrient and water absorption and utilization (*Zifčáková et al., 2016*). There are some differences in soil microbial community characteristics and composition in different distribution areas and seasons. In future research, the distribution area of the study can be expanded to the whole country, and annual dynamic changes of soil microbial community characteristics can be further studied. It is beneficial to understand the relationship between soil microorganisms and *C. oleifera* and its cultivation and management. At the same time, there are many unclassified and abundant microbial populations in the soil of *C. oleifera*. Perfecting the identification, classification, and naming of the dominant populations of unknown microorganisms are important to soil microbial community study. It would benefit to the development and utilization of microbial resources.

## CONCLUSIONS

In this study, we found that the nutrient conversion and carbon source utilization ability of the rhizosphere microflora of *C. oleifera* was superior to non-rhizosphere soil. When developing and utilizing the soil microorganisms of *C. oleifera*, the use of rhizosphere microorganisms can be given priority, especially in the growing season. In future management of forest land, it is possible to promote the growth of *C. oleifera* by controlling the carbon source species in soil and the soil physicochemical properties.

### Funding
This work was supported by the the National Key Research and Development Program of China (NO.2018YFD1000603; NO.2019YFD1001602). The funders had no role in study design, data collection and analysis, decision to publish, or preparation of the manuscript.

### Grant Disclosures
The following grant information was disclosed by the authors:
National Key Research and Development Program of China: NO.2018YFD1000603 and NO.2019YFD1001602.

### Competing Interests
The authors declare there are no competing interests.

### Author Contributions
- Peng Zhang conceived and designed the experiments, performed the experiments, analyzed the data, prepared figures and/or tables, authored or reviewed drafts of the paper, prof Xi revised the manuscript, and approved the final draft.
- Zhiyi Cui and Mengqing Guo performed the experiments, prepared figures and/or tables, and approved the final draft.
- Ruchun Xi conceived and designed the experiments, performed the experiments, analyzed the data, prepared figures and/or tables, authored or reviewed drafts of the paper, peng Zhang wrote the manuscript, and approved the final draft.

### Data Availability
Data is available at NCBI: PRJNA592248. Soil microbial community data is available as a Supplementary File.

### Supplemental Information
Supplemental information for this article can be found online at http://dx.doi.org/10.7717/peerj.9117#supplemental-information.

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
