# Peer review of "Characteristics of the soil microbial community in the forestland of Camellia oleifera"

_PeerJ, doi:10.7717/peerj.9117_

## Round 0.1 · original submission · Major Revisions

This paper needs a through revision. The introduction lacks a hypothesis
and the methods needs to be clear. In addition, the English needs a substantial revision as this is an international journal.

Reviewer 1 ·

Basic reporting

The English language of the manuscript is generally clear and understandable, but there are grammatical errors (e.g. “studies has”, l. 50) and incorrectly used words that cause some confusion (e.g. “soil oxidation” l. 42). The manuscript would benefit from proof-reading by a fluent first-language speaker.

The introduction is missing any literature background on soil microbial community analysis, including previous community characterization and how this relates to the present study.

The article is logically structured and easy to follow.

Figures are of high quality, although not all are well explained and some might be unnecessary (see further comments below).

It is not stated at what cultivation time point Figure 2 was measured (with respect to Figure 1).

To my knowledge, Figures 3 and 4 are rarefaction curves, not dilution curves.

Figures 7 and 8 Not clear if this is for rhizosphere soil or bulk soil.

The manuscript forms and appropriate, self-contained “unit of publication”

Experimental design

The manuscript fits well within the aims and scope of PeerJ.

The research question (determining the soil microbial community structure under C. oleifera) was reasonably clear, but the authors could better explain why this is relevant or important. A knowledge gap was not clearly identified.

There are several points of concern about the technical rigor with which the study was conducted, and the handling of the results:

1) I found it very surprising that so few OTUs were observed, especially for N4, N5 and N6 (Fig 1 and 2, line 290). Most remarkably, Figure 12 indicates that the fungal communities of N5 and N6 consist almost entirely of one genus! It would be good to know what steps were taken to check that no unusual artefacts were introduced during extraction, library prep or sequencing. Are the authors confident that these really reflect the diversity of these soil samples? This concern arises again in line 190-191, where it is reported that the 14 dominant non-rhizosphere bacterial species (OTUs?) accounted for 76% of total abundance – implying numerous OTUs with >5% of total abundance – a surprisingly simple community for soil.
2) Statistical significance in Table 1 seems incorrect. It is not stated what the uncertainty measures are (standard deviation or standard error? – this should be indicated). But even if these are standard deviations, it is not feasible that (for example) 5.2 +/- 0.29 and 5.33 +/- 1.14 could be significantly different (pH of non-rhizosphere soil in summer and autumn).
3) Figure 7 and 8 miss the point of a Venn diagram – the areas should be proportional to the values, otherwise the graphical representation has no meaning.
4) I doubt whether the PCA analysis actually provides useful information from this data, since the principle components seem to be mainly accounting for two outlier samples (I wonder whether these are N5 and N6?).

The Methods section is insufficient to allow for replication. In particular, it provides very little information on the procedures and conditions used for amplification and sequencing, and no information at all on bioinformatics analysis (e.g., which R packages, with what steps taken).

The Methods section should also provide a bit more information on the enzyme measurement procedures, and include a reference for the cellulase activity determination.

In line 81-83, some basic information on the sampling should be provided, at least including: sampling depth; separation of rhizosphere and non-rhizosphere soil; pooling of samples (this is clear from the data, but it is not clear in the Methods section whether the 15 trees are replicates are not); sieving.

I did not identify any ethical problems with the manuscript.

Validity of the findings

Concerns regarding the data and statistics have been noted above.

I was not able to open the background data files linked in the manuscript submission.

The manuscript does not mention the depositing of sequence data in an appropriate repository (e.g. NCBI), although this would be the norm for such a study.

Lines 29-30, and line 295, state that pH, available P and urease activity were “dominant influential factors for the soil microbial community”, but the data presented in the manuscript does not allow the direction of causality to be determined (even though, for pH and available P, it is already established in the wider literature).

Line 287-288 concludes that the “fertility” of the rhizosphere soil was superior, but “fertility” was never defined and it is unclear how this conclusion can be arrived at from the presented data.

Line 289-290 call the rhizosphere microbiota “superior”, but this value judgement is not defined in the manuscript.

The Biolog plates that were used to estimate functional diversity contain a range of very different chemical substances. It is not clear how these varied chemicals were grouped into just six categories (Figure 2, lines 245-247) and if this can actually say anything meaningful about function. For example, I assume lactose and glycogen were grouped under “Saccharides”, but how is it meaningful to consider these together? An ordination method might be more appropriate for analysing this data.

Figure 10 and lines 203-210 seemingly present data on functional genes in the community, but it is not explained where this data comes from or how it should be interpreted. The functions that are listed are all standard cellular housekeeping functions, so it is not clear what helpful information this gives about the soil community’s ecological function. Unless the authors can connect this to the interpretation of other data, I would suggest to remove this.

Line 222-223 conclude that microbial biomass and enzyme activities are good indicators of soil quality, but this cannot be concluded from the presented data, which did not contain any direct measures of soil quality (e.g. plant performance).

Lines 259-260 state that the dominant microbial groups are the basis of the soil’s function, but this is not demonstrated by the data, and it is not safe to assume that the most abundant taxa necessarily are the most important for function.

Lines 275-276 seem like speculation. This requires either supporting literature or a clearer link to the data.

Additional comments

Line 34-35 suggest an application of the findings, but this is not adequately explained in the text.

Line 38 “photo-contracted” should be “Photosynthetically fixed” or something similar.

Line 49 “buffering capacity” is not a good word to use here, due to its usual meaning in terms of pH buffering.

Are the sampled ecosystems really “forests” (which implies some level of natural function) or are they really plantations?

Line 74 Please provide a soil name in accordance with the World Reference Base, so that an international audience can easily understand the soil conditions.

Line 90 needs reference for alkali-hydrolysis diffusion method

Line 92 “resistance colorimetric method” unclear why resistance – not just a standard colorimetric / absorption method?

In the Results, I would encourage the authors not to repeat results in the text that is more easily understandable in tables or figures (e.g. lines 119-123), unless there is a clear reason to highlight one or two values of particular importance.

Line 163 – “airborne” not clear what this refers to

Line 171-172 The comparison here is not clear.

Various places, e.g. line 175, “OUT” should be “OTU”

The authors should carefully consider when it is appropriate to use the word “species” (in my view, when a species has been taxonomically defined), and when to use “OTU” (when it is based solely on a phylogenetic interpretation of the present data).

Line 198 It is unclear what “abundance” refers to here.

Lines 252-253 claim that rhizosphere soil should be more resistant to environmental soil, but this seems like speculation rather than an argument based on the presented data. This might be a missed opportunity, because the seasonal sample in this study might allow an analysis of which communities were more resistant and/or resilient to seasonal environmental changes.

Reviewer 2 ·

Basic reporting

There are many significant issues with this manuscript. For example,
1)No objectives and hypotheses are given.Hence there are no research questions.
2) Methods are not stated clearly and important information lacking e.g. conditions for molecular work and hence does not allow for meaningful replication.
3) In abstract, where the term, "greater than" is stated, there must be sound statistically evidence to support this. However, non was given.
4) Headings are not clear
5) Line 65-66 states.. "it is necessary to understand such information..... why is this so? 6) What is the significance to this study? How does it contribute to the gap in the literature and what is this gap? All these are very important questions that must be stated and addressed in a scientific study as this. However, these are lacking.

Experimental design

.

Validity of the findings

.

---

## Round 0.2 · Major Revisions

The mansucript has been revised, however some of the comments from the reviewers are still not properly addressed. In addition, the English expression still needs to be thoroughly revised. Currently there are still many obvious grammatical mistakes and unclear sentences. I suggest the authors seek an English editing service.

- The abstract and introduction still lack an explanation why this work is relevant or important for international community. What is the hypothesis?
- Fig 9, needs an explanation of what is the significance level
etc..

---

## Round 0.3 · Minor Revisions

The paper has been revised according to the reviewers' comments, however there are still some important issues that need to be addressed. The reviewer has kindly provided concrete suggestions that need to be incorporated in the revision.

Reviewer 1 ·

Basic reporting

- Literature background on soil microbial community analysis

In l. 64-71 the authors have added some commentary on laboratory culture studies, but it would be useful to have some context of the relevant microbiome research.

It is stated that “high throughput sequencing can reflect soil microbial community structure comprehensively and accurately” but this is not the case, as it suffers from numerous biases, e.g. (Větrovský and Baldrian, 2013), so this should be stated less strongly.

- Soil description according to the FAO World Reference Base

The authors have not described the soil in terms of FAO-WRB nomenclature, which would be very useful for readers from other regions of the world.

- There are various errors of English grammar, and the manuscript would benefit from proof-reading by a first-language speaker.

Experimental design

- Very few OTUs observed, especially for fungi in N5 and N6

The authors acknowledge that they were also surprised by these results. They try to explain the finding in terms of ecology (though not found in other replicates). However, it important first to demonstrate that the result is correct. It would be helpful to know what was done to make sure that these results were accurate. For example: Were the samples resequenced? Were mock communities properly sequenced with these samples? If the authors cannot provide assurance that this was not caused by experimental problems in sequencing, then this possibility should be acknowledged.

- Potential distortion of PCA by outlier communities

Related to the previous concern about OTUs, it would be helpful to know which two communities are the outliers in the PCAs in Fig. 5. In particular, if the two outlier fungal communities are N5 and N6 (as I suspect), it is important for the reader to know this, so that they can interpret the PCA appropriately.

Validity of the findings

- Incorrect significance calculations in Table 1

The specific example I noted in my previous comments has been changed, but the other values in the table appear to be unchanged. The authors should check all calculations in this table, since this was just an example of what seems to be a fundamental problem. As another example: pH for N in spring and winter is stated as significantly different: 5.56±0.35a and 5.33±1.14b.

Furthermore, the asterisks seem to indicate that rhizosphere and non-rhizosphere soil are different across the entire parameter, not pairwise statistical comparisons (for example, cellulase in summer: 0.47±0.04a* and 0.47±0.04a). I suspect this significance is derived from ANOVA, in which case it should not be presented for each data point. I suggest the authors instead show this significance in the soil column as R* if the rhizosphere is significantly different.

Finally, the footnote to the table in the original manuscript provided important information, but is missing from the revised manuscript.

- Causality versus correlation

The authors write in their response:

“As far as we are concerned, from the fig. 9 we have learned that pH, available P and urease activity were dominant influential factors for the soil microbial community.”

I disagree with this because “dominant influential factors” means that these are the causes and the microbial characteristics are the effects. Especially for urease activity, this is not obviously correct. One cannot determine causality from correlation, and Figure 9 presents correlations. Therefore I would recommend to soften this claim (abstract, line 28-29, 224-25) to rather use words like “associated with” which do not imply causality.

- Raw data
The authors assure that they have uploaded their sequence data to NCBI, but I can’t find other raw data in the revised submission.

Additional comments

Various positive changes have been made to the text and to the figures. There are, however, still points that I feel should be addressed to improve the manuscript.

---

## Round 0.4 · Minor Revisions

The paper has been revised accordingly. However the English should be revised. I suggest the authors seek an English editing service or ask an English speaker to edit their manuscript.

---

## Round 0.5 · accepted · Accept

The paper has been revised accordingly and it should be published now.